# Assessing Phenotypic Variability in Some Eastern European Insular Populations of the Climatic Relict *Ilex aquifolium* L.

**DOI:** 10.3390/plants11152022

**Published:** 2022-08-03

**Authors:** Ciprian Valentin Mihali, Constantin Marian Petrescu, Calin Flavius Ciolacu-Ladasiu, Endre Mathe, Cristina Popescu, Viviane Bota, Alexandru Eugeniu Mizeranschi, Daniela Elena Ilie, Radu Ionel Neamț, Violeta Turcus

**Affiliations:** 1Faculty of Medicine, “Vasile Goldiș” Western University from Arad, 310025 Arad, Romania; c.petrescu@landkreis-rastatt.de (C.M.P.); endre.mathe@agr.unideb.hu (E.M.); bota.viviane@student.uvvg.ro (V.B.); violeta.turcus@uvvg.ro (V.T.); 2“Aurel Ardelean” Institute of Life Sciences, “Vasile Goldiș” Western University from Arad, 310414 Arad, Romania; calinlad@hotmail.com; 3Research and Development Station for Bovine Arad, 310059 Arad, Romania; alexandru.mizeranschi@scdcbarad.ro (A.E.M.); daniela.ilie@scdcbarad.ro (D.E.I.); radu.neamt@scdcbarad.ro (R.I.N.); 4National Institute of Research and Development for Electrochemistry and Condensed Matter, 300569 Timişoara, Romania; 5Faculty of Agricultural and Food Sciences and Environmental Management, University of Debrecen, 4032 Debrecen, Hungary; 6Faculty of Pharmacy, “Vasile Goldiș” Western University of Arad, 310025 Arad, Romania; cristina.popesc@uvvg.ro; 7National Institute for Economic Research “Costin C. Kiritescu” of the Romanian Academy/Centre for Mountain Economy (CE-MONT), 725700 Suceava, Romania

**Keywords:** common holly, *Ilex aquifolium* L., morphometry, phytochemicals, rbcL, SE Europe, SEM

## Abstract

Through its natural or cultivated insular population distribution, *Ilex aquifolium* L. is a paramount species which is exceptionally suitable for studying phenotypic variability and plasticity through the assessment of morphological, physiological, biochemical and genomic features with respect to acclimation and/or adaptation efficiency. The current study is focused on four insular populations of *Ilex aquifolium* from Eastern Europe (i.e., in Romania, Hungary, Serbia and Bulgaria), and presents an initial evaluation of phenotypic variability in order to conclude our research on phylogenetic relationships and phytochemical profiles, including several descriptive and quantitative morphological traits. Taken together, the data from different methods in this paper indicate that the Bulgarian and Romanian populations can be distinguished from each other and from Serbian and Hungarian populations, while the latter show a higher level of resemblance with regards to their quantitative morphological traits. It is likely that these morphological traits are determined through some quantitative trait loci implicated in stress responses generated by light, temperature, soil water, soil fertility and salinity conditions that will need to be analysed in terms of their physiological, genomic and metabolomics traits in future studies.

## 1. Introduction

The *Ilex aquifolium* L. is an evergreen, dioecious and heterophyllous species with temperate region morphology characteristics that develops well on drained soils [1]. It is believed to originate from the Mediterranean area as the *Ilex-Taxus* complex [2], and is considered a tertiary relict species [3,4]. It is mentioned in The IUCN Red List of Threatened Species 2013 (www.iucnredlist.org, accessed on 24 April 2020). Currently, about 600 species from tropical to temperate regions are included into the *Ilex* genus (Aquafoliaceae) [5].

The *Ilex* L. species have been intensively studied using molecular approaches [6,7,8,9,10,11,12], eco/physiological techniques with molecular ecology and field studies [13,14,15,16,17,18,19], structural and ultrastructural characterization of different anatomical components [20] and phytoconstituents profiles [21,22,23]. Considering the European distribution area of *Ilex aquifolium*, in addition to studies carried out in Spain, Norwegian [24] and British Islands [6,16] field studies indicated its endangered nature.

Some data indicate that *Ilex aquifolium* is a native species of the eastern and central European region (Greece, Republic of Serbia, Bosnia and Herzegovina, Republic of North Macedonia, Montenegro, Bulgaria, Romania and Hungary) [15], while other reports consider it to be native to Greece, Bulgaria and the former Yugoslavian republics [25,26,27,28]. It was proposed as a foreign extant with an uncertain origin in Romania according to the IUCN Red List [29] and as a cultured species in Hungary [30], initially found in a few locations and then having spread throughout the country [31]. The existence of *Ilex aquifolium* on the territory of modern-day Romania was reported in 1893 as a part of the Arad county monography [32], and in the logs of Arad region flora [33]. They mention *Ilex aquifolium* in association with other species like *Fagus sylvatica* L., *Acer platanoides* L., *A. pseudoplatanus* L., *A. austriacum* Tratt., *Rhamnus tinctoria* L., together with different *Quercus* species. Nevertheless, it is quite puzzling that the same Simonkai, in a study published seven years earlier to the above mentioned Arad county monography, reported the presence of *Querqus ilex* L. without mentioning *Ilex aquifolium* [34]. Closer to the present, *Ilex aquifolium* was described in Romania [35] in 1993 as a protected species. The Romanian location for this species is in the “Dosul Laurului” natural reservation (0.33 km^2^), founded in 1999 and situated in the Codru-Moma Mountains, Zimbru population, Gura-Hont commune, Arad County, with GPS coordinates 46°23′55.5″ N, 22°22′50.9″ E, according to the UNEP-WCMC and IUCN in 1999 [36,37].

Regarding its habitat and plant association, the RO population of *Ilex aquifolium* consists mostly of trees and, to a lesser extent, shrubs, with heights of approximately 3–20 m, while in HU, 1–5-m-tall shrubs and subshrubs can be found (Appendix A). In SR, 3–5-m-tall shrubs were observed, while, interestingly, the BG population consists only of a few subshrubs with heights less than 1.5 m (Appendix A). The RO and SR populations are found in mixed forests of deciduous and sempervirens wet plants, often in association with species of the Betulaceae family and Quercus genus, although in SR, some species of Pinaceae were also present. In the case of HU, the *Ilex aquifolium* population was found in association with different plant species (such as *Taxus baccata* L., *Sophora japonica* (L.) Schott., *Sequoiadendron giganteum* (Lindl.) J. Buchh., *Ginkgo biloba* L., *Taxodium distichum* (L.) Rich.) [38], which was not surprising, given the anthropic nature of the studied arboreal park. The BG population, being part of the mountainous zone flora, was located at the bottom of a valley which was partially covered by vegetable fodder from different species of Pinaceae (Appendix A).

Molecular studies carried out on 108 *Ilex* species revealed the diversification history of the *Ilex* genus, as inferred from analyses of the markers for its nuclear (ITS, nepGS) and plastid sequences (rbcL, trnL-F, atpB-rbcL), [4]. Plant DNA barcoding regions are commonly used in the genus, species, ssp. identification because they provide a rapid, accurate, and automatable mode of identification. They are used as single entities (i.e—RbcL) or in tandem (i.e—rbcL-matK, trnH-psbA, trnL-F) [39]. In our study, for gene validation at a genus/species level, we propose the rbcL barcoding plastid region, given that the use of rbcL gene sequences effectively identifies most samples [40].

Most morphological studies of *Ilex aquifolium* have been related to the leaf ultrastructure [20], while some aspects of root development were elucidated using micropropagation techniques [41]. Interestingly, the heterophyllous leaf morphology was correlated with DNA methylation, i.e., the DNA in the nonprickly leaves seemed to be more methylated than that in the prickly ones [9]. The authors of this study also support the role of epigenetic modifications as well as genetic variation in inducing phenotypic variations among natural plant populations. The *Ilex aquifolium* species has been well described previously from an ecological and botanical point of view, but only briefly from the point of view of its phytochemical compounds. Based on the origin of the main metabolic pathways and on previous studies, four classes of dominant compounds can be characterized: alkaloids, terpenes, sterols and phenolic derivatives [21,42,43,44].

Several analytical chemistry studies have been carried out on the leaves, fruits or seeds of *Ilex aquifolium*, and various phytochemicals have been identified, like terpenoids, sterols, cyanogenic glucosides, anthocyanins, flavonoids, saponins and phenylacetic acid derivates [21]. Studies that compared the specific phytochemical profile of *Ilex aquiafolium* leaves to those of other species like *Ilex paraguariensis* A.St.-Hil., ‘Yerba Mate’, *Ilex aquifolium*, ‘Argentea Mariginata’ and *Ilex × meserveae* Kath. Meser., ‘Blue Angel’ revealed significant differences [22]. Similarly, significant differences were evident in the phytochemical profiles of the leaves of other *Ilex* species used for tea preparation [23], suggesting that such studies could be informative for the presently studied *Ilex* species.

The aim of the present study was to perform a comparative analysis of the descriptive/quantitative morphological traits, phytochemical profiling and DNA barcoding validation of individuals that belong to insular populations of *Ilex aquifolium* located in Romania, Hungary, Serbia and Bulgaria. In the case of the native BG and SR populations and the cultivated HU population, these have been previously recognized, while the origin of the RO population is awaiting scientific confirmation (neobiota—a Hungarian or Serbian origin or a native population?). As such, our goal in this study was to set the grounds for a long-term study on the above mentioned four populations in order to assess their adaptability and acclimation potential with respect to the evolving nature of the climatic conditions in Eastern Europe.

## 2. Results

### 2.1. Molecular Analysis

In order to assess the validation and the relationship between the studied RO, HU, SR and BG samples, we focused on the rbcL gene, which is considered a generic plastid marker [45]. We used a multiple alignment test for all four sequences which generated similarity between the sequences in RO, SR and BG, respectively, a difference at the SNP (single-nucleotide polymorphism) level between all these three sequences and the HU sequence. We compiled a neighbor-joining tree where the data clearly indicated this aspect, i.e., the difference between the RO, SR, BG sequences and the HU sequence.

The NCBI query results yielded around 100 *Ilex* genera matches, while the Clustal W multiple alignments suggested that the assessed RO, HU, SR and BG samples belonged to the *Ilex aquifolium* specific clade. Furthermore, we compiled a bootstrap consensus tree using 20 NCBI specific *Ilex aquifolium* sequences, together with our four population-specific rbcL sequences, as can be seen in Figure 1. Our data, depicted in a phylogenetic tree, clearly confirmed that the RO, HU, SR and BG samples belonged to the *Ilex aquifolium* species, although some subclades were apparent. The RO and SR populations appeared to be more closely related, being included on the same branch, whilst the BG population formed a separate branch, having a common node with the RO/SR branch. The HU population was the most phylogenetically distanced from the others.

### 2.2. Leaf-Specific Phytochemical Profiles

Leaf specific hydroethanolic extracts were prepared and analyzed by UHPLC-ESI-MS. The main phytochemicals were amino acids, carboxylic acid, polyphenols, vitamins, flavonoids, coumarin derivatives, saponins, terpenoids and carbaldehyde derivatives. Fifty-five phytochemicals were identified in RO/HU/SR and 46 in BG, as shown in Table 1:

The data indicate that there are more relevant similarities between the RO, HU and SR leaf specific phytochemical profiles than the BG, which seemed to contain fewer compounds. In this respect, certain flavonoids (eriodictyol-O-hexoside, isoquercitrin, kaempferol-O-hexoside, quercetin-3-O-rutinoside-7-O-glucoside, quercetin-O-hexoside isomer, reinutrin), carboxylic acids (12-Oxo phytodienoic acid, hexadecanedioic acid), polyphenols (feruloylquinic acid isomer 1 and 2), a terpenoid (uvaol) and a vitamin like pyridoxine were not detected in the BG samples. Polyphenols like feruloylquinic acid isomer 1 and 2, together with the Matesaponin 3, were absent from the RO samples but present in all other studied leaves. Taken together, similar leaf phytochemical profiles were observed for all four *Ilex aquifolium* populations, although some differences were visible. Considering the total number of identified phytochemicals which are known to be present in *Ilex aquifolium*, surprisingly, the BG leaves were missing 13 compounds, while for the HU, RO and SR samples, only four compounds were absent.

In total, 46 phytochemicals were identified in BG and 55 in the HU, RO and SR leaves (Figure 2). See Appendix A–Total Ion Chromatograms *Ilex*—Appendix A; Phytochemicals identified in the *Ilex* extracts—Appendix A.

### 2.3. Morphological Traits by SEM Descriptive Analysis

#### 2.3.1. Leaf

All *Ilex aquifolium* leaf samples had their adaxial and abaxial foliar surfaces strongly cutinized and thickened (Figure 3a–h). The adaxial surface presented fine cuticle striations that were markedly visible in the BG, RO and SR samples (Figure 3a,c,d arrow heads), but less evident in the HU samples (Figure 3b). No trichomes or stomata were found on the adaxial surface of the leaves. In the case of the BG samples, on the adaxial leaf epidermis, pentagonal or hexagonal shaped cells were observed (Figure 3d, thick arrows). Interestingly, fungal hives could be seen on the adaxial and abaxial surfaces of both the HU and RO samples (Figure 3a,b,e,f, thin arrow indicators).

The abaxial epidermis of the analyzed leaves exhibited cuticular striations to different extents, such that the phenotypic strength of each sample could be described on a scale from more to less pronounced. In case of the HU and SR leaves, the cuticular striations were more evident and had a higher density (Figure 3f,g). The specific cuticular striations on the RO and BG leaves were thinner and of a lower density (Figure 3e,h). On the abaxial surface of all leaves, there were ranunculaceous type stomata with the stomatal apparatuses having a round-oval shape and the orientation of longitudinal aperture being randomly distributed (Figure 3e–h).

The inner wall thickness of the stomatal pore appeared to vary across the samples, i.e., in the HU and SR samples, it was more prominent, while in the RO and BG samples, it looked relatively thinner. The morphology of the cuticular areas surrounding the stomata were more evident in the BG leaves, as the cuticular striations featured the lowest density (Figure 3h).

On the abaxial side of all four location-specific *Ilex aquifolium* leaves, we were able to observe some giant or D-type stomata apparatuses (Figure 4a–d, thin/thick arrows), as has been previously reported for *Ilex crenata* Thunb. ssp. convexa. The D-type stomata were surrounded by several normal peripheral stomata (Figure 4b, thin arrows) in a radiant orientation, delineating a relatively circular space (Figure 4a–d—circle).

We assessed prickly leaves and the lateral thorn area thereof was SEM analyzed (Figure 4e–h). Once again, the phenotypic strength seemed to vary based on the location. The lateral thorn area specific cuticle showed more pronounced punctuation for BG (Figure 4h), while slight longitudinal cuticle striations were observed for HU and SR (Figure 4f,g). Weaker but still visible punctuations were apparent in the RO samples (Figure 4e).

#### 2.3.2. STEM Surface and Trichomes

The stem-specific phenotypes were SEM analyzed further in the case of one-year old shoots from samples collected from all four locations, and several phenotypic features were identified (Figure 5a–d). Similar to the RO and HU leaf surfaces, the presence of fungal hives that has developed deposit structures was observed on the stems of the RO and HU samples (Figure 5a,b, thin arrows).

The existence of numerous cuticle deposits covering the stem-specific insertion base of the stem trichomes on the epidermal surface was more evident in the HU and RO samples (Figure 5e,f) compared to the SR and BG samples.

Looking at other phenotypic aspects for stem trichomes from a tilted angle in our SEM analyses, we observed different triangular shapes, and a specific shape was observed in every location; see Figure 5i. The prominent stem trichome shape looked like an isosceles triangle with a low base for RO, an acute-angled isosceles triangle for HU, an isosceles triangle with a rounded upper apex in SR, and an isosceles triangle with a larger base for the BG samples; see Figure 5i. Moreover, assessing the surfaces of stem-specific trichomes, verrucous structures were seen in SR (Figure 5g, thick arrow) and moderate verrucous surfaces for RO and HU, (Figure 5e,f, thick arrow), while for the BG samples, the verrucous surface was underrepresented or totally absent; see Figure 5h.

Regarding the trichome density on stem surfaces, the phenotypic strength varied to different extents, such that the RO samples seemed to show the highest density, followed by HU and SR, while the BG stems featured the lowest trichome density; see Figure 5a–d.

### 2.4. Quantification and Statistical Analysis of Leaf- and Stem-Specific Morphological Traits

Analyses were performed for the following morphological traits (MTs), which were measured as described in Section 4.4: leaf stomata distance—SD; leaf D-type stomata distance—DTS; leaf stomata length—SL; leaf stomata width—SW; stem trichome length—STL; leaf stomata frequency—SF. In the context of our statistical analysis, the assessed MT-specific data were used to compute the basic statistical parameters (mean, median, sd); see Table 2.

In accordance with our datasets, we compiled box plot diagrams for all of the aforementioned MT. The box plots depict the median related variation interval (true distribution) of the individual measurements, allowing a visual comparison to be made of MT-specific datasets for all four local populations; see Figure 6.

MTs correlations were as follows: in SD, similarities were found between the HU and SR populations and differences among them compared with the RO and BG populations; in DTS, there were similarities between the HU and SR populations; in SW, there were differences among RO and BG populations; in SL, there were differences among RO and BG populations; in STL, similar values with an increasing tendency were observed in HU, SR, BG and RO; in SF, similarities among HU and SR and more negative values in RO and BG were observed. For STF, none of the populations revealed similarities among the four locations.

Regarding the distribution of values specific to SW and SL, they looked relatively similar in the case of the RO, HU and BG populations, while SR showed a certain difference related to the range of variability of the two traits. Based on the median values, there was a similarity between the populations in HU and SR in most of the analyzed MTs, while in the populations from RO and BG, the values were different from those of the first subgroup (HU-SR) and also between each-other.

Together with basic statistical parameters, we used the nonparametric Kruskal-Wallis test, followed by Dunn’s test for pairwise comparisons, in order to check for significant differences between the median values of each of the seven MTs across the four locations. The results are summarized in Table 3.

A *p*-value cut-off of 0.05 was used to assess significance, and the Benjamini-Hochberg FDR approach was applied for *p*-value adjustment for multiple comparisons in the Dunn’s test.

Significant differences among countries varied among the MTs. For SD, for example, significant differences were as follows: RO > HU; RO > SR; HU < BG and SR < BG. For DTS, none of the comparisons revealed significant differences among the four locations. For SW, significant differences were as follows: RO < HU/SR; SR > BG; no significant differences among RO/BG and HU-SR. In SL, RO < HU/SR; HU/SR > BG and no significant differences among RO—BG and SR—HU. For STL, significant differences were as follows: RO > HU/SR/BG and no significant differences among HU—SR/BG and SR—BG. For SF, significant differences were as follows: HU > Bg, SR > BG and RO > BG and no significant differences among RO—HU/SR and HU—SR. Finally, for MT and STF, significant differences were as follows: RO > HU; RO > BG and SR > BG.

Following the nonparametric Kruskal-Wallis test and by Dunn’s test data for pairwise comparisons of each of the seven MTs across the four locations, the results showed that there were no significant differences among the HU and SR populations in any of the studied MTs; a grouping tendency of all the four populations in three sub-groups was observed based on the MT analysis, i.e., a first sub-group from the RO population, a second from BG population and another from the HU and SR populations. This agrees with the primary statistical data presented in the boxplots (Figure 6).

In order to reduce the dimensionality of our dataset and to identify relevant MTs for every population, we applied PCA, Figure 7. Looking at the positioning of country-specific values along the PC1 and PC2 axes, in which three major distribution areas can be identified. The HU and SR populations are overlaid, while the BG and RO populations have distinguished PCA bi-plot positions.

Clearly, the HU and SR populations feature more significant variability for SL and SW, while less pronounced variability was evident for SF. These observations suggest that the SL and SW, together with SF, are determinative quantitative traits, allowing the populations to adapt to specific environmental conditions. It was also noticeable that SD and DTS seemed to reduce the variability in the HU or SR populations.

The first two principal components (PCs) accounted for more than 60% of the total variance among the 40 samples, with PC1 and PC2 explaining 43% and 21.9%, respectively. When plotting the first two PCs (Figure 7) and coloring the data points according to their location, the data points for individuals originating from the same location (country) were observed to cluster together. Furthermore, the HU and SR clusters overlapped, forming a distinct cluster of RO and BG, suggesting that individuals from HU and SR were relatively similar in terms of the seven MTs included in the study, based on the >60% variability captured by the first two PCs. Figure 7 also shows the seven PCA loadings corresponding to each MT. It can be seen that the STL and STF loadings pointed toward the RO cluster, which suggests that the corresponding MTs had larger values, on average, for the RO individuals than for those from the other three locations. A similar trend can be observed for SL and SW, which were oriented toward the HU and SR clusters. Conversely, the SF PCA loading was directed opposite to the BG cluster, suggesting that BG individuals had overall smaller values for this MT than individuals from the other three locations. A similar pattern was observed for DTS and SD, which were directed opposite to the HU and SR clusters.

A Mantel test based on Pearson’s product-moment correlation was performed to see if there was a significant correlation between the distance matrices of dendrograms corresponding to the phytochemical data (MTs Figure 8) and bioclim variables (phytochemicals/MTs, Appendix A). The results did not reveal a significant correlation between these distance matrices.

## 3. Discussion

In the present study, we performed a comparative analysis of some molecular, morphological and chemical features of four insular populations of the *Ilex aquifolium* species located in the Balkan and Carpatho-Pannonian biogeographic regions spanning eastern European countries like RO, HU, SR and BG.

*Ilex aquifolium* is considered a climatic relict species, but its tertiary or postglacial nature is still debated [46,47]. The tertiary relicts have been proposed to withdraw from their initial geographical distribution with the onset of a drier and cooler climate during the late Tertiary and early Quaternary, while postglacial relicts survived to the present day in regions with warmer climates. It has also been suggested that the majority of European climatic relicts could derive from Mediterranean refugia populations that ultimately underwent a northward expansion in the post-glacial periods [48]. Interestingly, the existence of a Balkan-placed *Ilex aquifolium* refugium was assumed upon analyzing populations from Croatia (45.87 latitude and 15.95 longitude) and Greece (40.56 latitude and 23.73 longitude), without considering the northernmost RO, HU, SR and BG populations. The existence of isolated populations of *Ilex aquifolium* in RO, HU, SR and BG raises some intriguing questions regarding their phylogeny, reaction norm, ecophysiology and adaptation to the climatic conditions in the Balkan peninsula [49].

**Molecular analyses indicate that all four populations belong to the *Ilex aquifolium* species.** Similar recent studies have used the rbcl gene with high identification rates within the *Ilex* genus [50]. In accordance with our results, the HU population seems to be fairly distanced from the analyzed BG and RO and SR populations, while also considering other samples from countries such as Italy (IT), United Kingdom (UK) and Switzerland (SW); see Figure 1. Moreover, the HU population cannot be considered native, as it was introduced into the collection of shrubs and trees in the Szarvasi arboretum in the past, although records referring to the country of origin and inclusion time were lost. Therefore, the origin of the HU population remains an open question and, to further strengthen the phylogeny of the reported populations, we intend to assess other plastidic (rbcL, matK, rpoB, rpoC1, microsattelites) and nuclear (nrITS) types of molecular genetic markers [11,12] in future studies.

**The phytochemical profile data obtained in the present study identified but also detected new phytochemical fractions in the chemical composition of *Ilex aquifolium* leaf**. Based on the analysis of phytochemical fractions, as a first important observation, we note that the most phytochemical compounds identified from *Ilex aquifolium* leaves in this study had been partially characterized in the literature only in *Ilex paraguariensis*, such as: naringerin [51], kaempferol and quercetin derivatives [51], hexanoic acid [52], isoquercitrin [53], feruloylquinic acid derivative and cis-O-(4-Coumaroyl) quinic acid derivative, trans-3-O-(4-Coumaroyl) quinic acid derivatives [54], chryptochlorogenic acid [55], neochlorogenic acid [56], mateglycoside B and matenoside A [57], matesaponins [58], abscisic acid [59], nicotinic acid and riboflavin [60]. A smaller number of phytochemical fractions are described in other species belonging to the *Ilex* genus: *Ilex pubescens* Hook. and Arn.—uvaol [61], *Ilex macropoda* Miq.—betulinic acid [62], *Ilex guayusa* Loes.—amino acids [63], *Ilex cornuta* Lindl.—phenolic carboxylic acid derivatives [64], *Ilex integra* Thunb.—carboxylic acid [65], *Ilex kaushue* S.Y.Hu—di-O-caffeoylquinic acid isomer derivatives [66], *Ilex* × meserveae S. Y.Hu—matesaponin 4 [67], *Ilex* sp.—rutin, isorhamnetin derivatives, feruloylquinic acid isomers and quinic acid—[22,43]. There were similarities between the phytochemical fractions in the present study and those presented in other recent studies for oleanolic and ursolic acids in *Ilex aquifolium* [68] and citric acid and hexadecanedioic acid [69].

A few phytochemical compounds were identified for the first time, such as: 1-benzofuranecarbaldehyde; indole-4-carbaldehyde; flavonoids [eriodictyol-O-hexoside, homoeriodictyol (3′-Methoxy-4′, 5,7-trihydroxyflavanone)]; coumarin—12-hydroxyjasmonic acid-12-O-glucoside like and tuberonic acid glucoside; 12-Oxo phytodienoic acid; 12-hydroxyjasmonic acid-12-O-glucoside like tuberonic acid glucoside; trans-melilotoside (trans-glucosyl-2-hydroxycinnamate); adenine and pyridoxine.

Interestingly, when the phytochemical profile of leaves was assessed, once again, the BG population featured significant qualitative differences compared to the HU, RO and SR samples (Table 1, Figure 2).

It has been shown that the synthesis of phytochemicals during the secondary metabolism of plants is geo-climate-dependent and could be correlated with a wide range of environmental factors such as light, temperature, soil water, soil fertility and salinity [70]. Modifying a single environmental parameter may cause phytochemical profile changes, even if other environmental factors remain constant. Table 4 [71] shows the values of some environmental factors specific for the RO, HU, SR and BG sampling areas. On this basis, it can be inferred that the BG location, having the highest altitude of all the sampling areas (1143 m), would feature the lowest values for the photoperiod, cloudy/sunny days/month and day and night temperatures. Such combinations of environmental factors could serve as plausible explanations for the lower number of phytochemicals present in BG leaf, i.e., 46, as compared to the 55 seen in the HU, RO and SR leaves.

Mostly similar flavonoid profiles were identified in all four locations, although their number appeared significantly lower in the BG leaves (Figure 2). It has been demonstrated that the presence of less accessible ground water and/or drought could cause phytochemical profile changes by increasing the flavonoid content in the leaves of pea plants [72] and *Amaranthus* L. [73]. It is therefore possible that the lower flavonoid presence observed in the BG leaves could be a consequence of a higher soil water content, as opposed to the higher flavonoid content in the HU, RO and SR leaves, that may have been due to stress inducers like a much lower water regime. It is also interesting that not just the number, but also the total contents of flavonoids and other primary metabolites were significantly lower for BG compared to the other samples. Recently, it was reported that drought could induce important transcription and metabolic adjustments, together with morphological (stomatal closure) and physiological changes in the leaves of *Ilex paraguariensis* [62].

There were higher degrees of similarity between the HU and SR phytochemical profiles regarding the presence of amino acids, carboxylic acid, coumarin and terpenoids, and differences in the presence of flavonoids, polyphenols and saponins. The polyphenolic and flavonoid profiles compound numbers were greater in comparison with those for terpenoids, which were decreased in all four locations. These results could be a consequence of an increased CO_2_ concentration [74,75]. Therefore, it seems likely that the samples from the four locations belong to two phytochemical profiles, with one represented by the individuals from BG and the second by RO, HU and SR, as the hierarchical clustering of the phytochemical data also showed; see Figure 8.

Taken together, the molecular validation and phytochemical profile analyses suggest that the BG population is relatively distinguishable from the others, while cultivated HU individuals showed a similar phytochemical profile to the native RO and SR populations.

In order to evaluate the distance isolation (IBD) and the environment isolation (IBE), the Mantel test was used. The results did not reveal the existence of a significant correlations between phytochemicals/MTs and climate/geographical data. Similar studies [76] have shown that chemical and morphological variability are not always associated with geographical and environmental variables. For a better characterization of interpopulation correlations, similar studies of (IBD) and (IBE) were performed using bioclimatic variables and genetic distances [77].


**Descriptive morphological traits and MT set BG/RO apart from HU/SR**


The gradual decline of water, along with cooling over the last 15 million years, has led to a change in the European plant profile. Species from previously widespread wet temperate tertiary forests have found refuge in the Balkan Peninsula as well as in Turkey [78,79]. Studies on the diversity of trees in the Balkans and the Carpathians have been carried out previously. For example, at least two oaks lineages (*Quercus* ssp.) have been identified: one located east of the Carpathians and another in the south [80,81]. It has been proposed that the thermal relict features of plant species in the Northern hemisphere are related to their isolated occurrence at great distances [47]. The geographical positions of the studied insular but natural populations of *Ilex aquifolium* (with exception of HU population) are relatively distanced from each other, both latitudinal and longitudinally, with north–south distances of 150–300 km. Similar studies on the variation of MTs at a species level, in different populations from the Balkan Peninsula, were performed on the service tree (*Sorbus domestica*) L., a rare and endangered species [82]. The results revealed significant variations within and between the analyzed populations regarding the morphological features of the leaves in the studied populations from the west, the center of the Balkan Peninsula and South-Central Europe. A comparative analysis of their morphological features could shed some light on the phenotypic variability across the insular populations, and, as such, could be considered an initial step in a lengthy attempt to monitor adaptation. The cultivated HU population is the result of anthropogenic intervention, and it is geographically relatively close to the natural RO population, so a longer timescale comparison thereof could reveal other aspects related to adaptive efficiency.

Studies on *Jovibarba heuffelii* (Schott) by Á. Löve and D. Löve showed that some environmental and bioclimatic factors (such as elevation, habitat, slope, photoperiod, humidity, etc.) can influence the morphology of a species, thereby guiding interpopulation differentiation [83].

Recent experiments have shown that the cuticle thickness, stomatal density, shape and density of trichomes are involved in adaptive mechanisms of the leaf to reduce water loss, playing an important role in protecting the cuticles by integrating flavonoid compounds as a response to the light and UV intensity [84,85].

The SEM analysis of some descriptive morphological traits specific to leaf, stem and thorn cuticle regions showed aspects that set apart or bring together the assessed populations of *Ilex aquifolium*. Morphological resemblances were observed between the HU and SR populations by the presence of a prominent follicular cuticle (Figure 3f,g). In the RO and HU populations, fungus hives were found at the level of the stem surface (Figure 5a,b,e,f), an aspect already reported [20] in the case of *Quercus ilex* from Central Spain. The morphology of trichomes appeared similar in terms of shape between the RO and BG populations, having the form of an isosceles triangle with a lower base in RO and a higher base in the BG population (Figure 5i).

The cuticle aspect of the foliage thorn connects the RO and BG populations, both of which showed cuticulary punctuations (Figure 4e,h), while the SR and HU populations had longitudinal strips (Figure 4f,g). The stem-specific trichome density varied across populations, with the RO samples showing the highest and the BG the lowest density (Figure 5a,d). Trichomes have been shown to protect plants from excess transpiration, high temperature, radiation, UV light and herbivore attack [86]. Therefore, it seems reasonable to presume that the differences in trichome density might be evidence of some adaptive responses to environmental changes in addition to genetic and vegetative phase-specific control, as demonstrated in the case of *Arabidopsis thaliana* [87]. Taken together, our observations of some morphological traits suggest a certain level of resemblance between the BG-RO and HU-SR populations, even though differences were also apparent among the studied four populations. In some cases, the morphological traits seemed to vary across the four populations. This was the case for the surface of stem-specific trichomes, where the BG stems were almost missing verrucous structures, while the RO and HU samples showed good representations thereof, and many such structures were visible on SR samples (Figure 4 and Figure 5). Assessing the descriptive morphological traits did not allow us to clearly define each population, and without recording numerical data, it was impossible to look for correlations among traits and/or environmental influences.


**MTs with continuous distribution ranges are essential for population-specific phenotypic variability**


A box plot analysis revealed the continuous distribution of the assessed MT-specific values in an interval range related to the median value of each trait and location (Figure 6). The higher degree of variability seen for DTS and SF in all locations could be attributed to local environmental factors like light intensity and CO_2_ concentration, as suggested for the stomatal development of *Arabidodpsis thaliana* (L.) Heynh., [88]. STF had a lower level of variability for HU/SR/BG and relatively higher level of variability in the RO population. It is possible that the reason for such a population-specific distribution is related to the local hydric regimes (see Table 4), as direct correlation was found between stem trichome development/frequency and a deficit in air moisture or water regime (as present at higher altitudes) [89,90].

Meanwhile, for the RO population, STL and STF were more determinative, although DTS could also be invoked, but to a lesser extent. Therefore, it seems likely that both STL and STF could be the major environmentally influenced quantitative traits in the RO population.

Interestingly, the BG population-specific cluster comprising the variability of all MT seemed not to feature any MTs exerting an increasing or reducing effect on variability. Such an observation would indicate that the BG population is fairly stable, at least in the case of the analyzed MTs, which also means that the climatic conditions do not fluctuate excessively. The stability of the BG population is further substantiated by the scatterplot data, indicating six, the largest number of MT correlations among all the studied populations.

## 4. Materials and Methods

### 4.1. Taxon Sampling

All plant specimens used in this study belonged to the *Ilex aquifolium* species. At each location (see Figure 9), 10 samples of one-year old shoots (one shoots/plant) from ten female plants were collected at around 1–2 pm from October to November 2017.

The specific geo-climatic factors for the *Ilex aquifolium* harvesting areas are summarized in Table 4. Samples comprising 10 shoots (in each place from all locations) were preferred due to the fact that the Bulgarian population was poorly represented as individuals; see Appendix A. As a consequence, we decided not to use traits such as leaf size and serration number to represent morphological differences, even though these seem to be widely used, especially in evaluations of ecological features. The collected samples were wrapped in aluminum foil (to avoid contamination) and stored at −80 °C prior to any examinations.

### 4.2. DNA Extraction, PCR Amplification and Sequencing

The total DNA obtained from leaves for the analyzed individuals were used to amplify and sequence the rbcL region. With the generated sequence data, NCBI blast searches were carried out.

Two prickly leaves from every collected shoot per sampling area were pulled together, and the total genomic DNA was extracted using DNeasy Plant Mini Kit (Qiagen, 19300 Germantown Road Germantown, MD 20874) following the manufacturer’s protocol. The DNA concentration was quantified by a spectrophotometric method using a NanoDrop-2000 (Thermo Fisher Scientific Inc., Waltham, MA, USA). The primers and methodology for PCR amplification conditions were as follows: rbcLa fw 5′-ATGTCACCACAAACAGAGACTAAAGC-3′ and rbcLa rev 5′-GTAAAATCAAGTCCACCRCG-3′, mastermix DreamTaqPCR (Thermo) with a PCR reaction volume of 25 μL/sample; PCR conditions: 98 °C 0.45″, 98 °C 0.15″, 59 °C 0.30″, 72 °C 0.40″, 35 cycles [91]. A modified protocol amplification was used in the case of Romanian samples by addition of 1 μL non-acetylated BSA and 1 μL undiluted Tween 20 to PCR volume reaction/sample as the extra reagents [92]. The PCR products were sequenced by an outsourced sequencing service [93]. The GenBank accession numbers released for nucleotide sequences are: specimen Seq1Ro MK300212, Seq2Bg MK300213, Seq3Hu MK300214, Seq4Sr MK300215. Post-sequenced processes, adjustment, assembly, NCBI blast search [94] and multiple alignments were performed using MClustalW/Omega [95], BioEdit Sequence Alignment Editor v.7.0.5.3 [96]. Phylogenetic tree depictions were created using MEGA v.6. Phylogenetic reconstructions employed the maximum likelihood method based on the model proposed by Tamura and Nei [97].

### 4.3. Phytochemicals Analysis

*Ilex aquifolium* fresh leaves were frozen in liquid nitrogen and ground to obtain a fine powder. The Nihal [98] extraction method was used with slight modifications. Briefly, 5 g of leaf powder were mixed with 50 mL 60% methanol and extracted for 24 h at room temperature using a rotary shaker. The containers holding the extract were wrapped in aluminum foil in order to protect the extract from sunlight. After extraction, the mixtures were filtered using vacuum filtration. A Dionex Ultimate 3000RS UHPLC system equipped with Thermo Accucore C18 column, 100/2.1 with a particle size of 2.6 μm was coupled to a Thermo Q Exactive Orbitrap mass spectrometer equipped with an electrospray ionization source (ESI); the measurement accuracy was within 5 ppm. All other features of the analysis were carried out as described in previous experiments [99]. The capillary temperature of the mass spectrometer was 320 °C, while the spray voltage settings were 4.0 kV/positive mode and 3.8 kV/negative mode. The scanned mass interval was 100–1000 m/z with a resolution of 35,000 for MS and 17,500 for MS/MS, so the latter was operated at a collision energy of 40NCE. Depending on the type of ionization, specific eluent mixes were applied for the UHPLC separation. A combination of eluent A (500 mL water containing 10 mL acetonitrile, 0.5 mL formic acid, and 2.5 mM ammonium formate) and eluent B (500 mL acetonitrile containing 10 mL water, 0.5 mL formic acid and 2.5 mM ammonium formate) was used for the positive ionization. In the negative ionization mode, the elution mix comprised eluent A (500 mL water containing 10 mL acetonitrile and 2.5 mM ammonium acetate) and eluent B (500 mL acetonitrile containing 10 mL water and 2.5 mM ammonium acetate). A 200-μL/min flow rate was applied in combination with the same gradient elution program for both positive and negative ionization-specific determinations (0–1 min, 95% A, 1–22 min, 20% A; 22–24 min, 20% A; 24–26 min, 95% A; 26–40 min, 95% A). Lower than 5 ppm differences between the measured and calculated molecular ion masses were considered. Finally, 5 μL of each Ilex extract was injected at every run.

### 4.4. Morphology Analysis and Morphometry

For our morphological analyses, a Quanta 250 FEI scanning electron microscope (SEM) was used. The samples were coated with gold using an AutoAgar sputter-coater. Three gold deposit layers (4 nm thick) in cycles of 10 s were applied to every sample surface. The samples were examined under vacuum using a secondary-electron detector. Measurements were carried out for the following MTs: leaf stomata distance—SD; leaf D-type stomata distance—DTS; leaf stomata length—SL; leaf stomata width—SW; stem trichome length—STL; leaf stomata frequency on approximately 1.5 mm^2^—SF and stem trichome frequency on approximatively 3 mm^2^ each—STF, as shown in Figure 10.

With the exception of STF, a trait assessed on stem cuticle surface, all the other MTs were studied on the lower/abaxial surface of prickly leaves situated on one-year-old shoots of female individuals. From the mid-zone of every shoot, a leaf was selected and further processed for microscopic examinations.

From each location (RO, HU, SR and BG), a 10 individuals were studied, making 40 individuals in total. For each individual, seven morphological traits (MTs) were assessed, from which five (SD, SW, SL, STL and DTS) were represented as mean values per character per individual and two (SF and STF) were expressed as absolute values obtained by counts per unit of surface (SF per 1.5 mm^2^ and STF per 3 mm^2^) per individual. Mean values were obtained from 100 measurements, except for DTS, where the averages were calculated from 10 measurements/individual, because the frequency of appearance of these structures was comparatively low.

All SEM measurements were performed with the IS Scandium Image Analysis software [100].


**Statistical analysis**


Due to the small number of data points per location and the deviations within the data from normal distribution (results not shown), the assumptions for running ANOVA were not met.

We chose to assess the same set of seven quantitative MTs for every studied population, and a great deal of primary data were recorded and statistically analyzed. Right after computing the basic descriptive statistical parameters (see Figure 6), we chose to use the nonparametric Kruskal-Wallis test, followed by Dunn’s test for pairwise comparisons, in order to check for significant differences between the median values of each of the MTs across the four locations (RO, HU, SR and BG), Table 3.

Statistical analyses were performed on the 280 datapoints outlined above, for the 10 individuals for each of the four locations, characterizing seven MTs in each individual. Microsoft Excel 2019 and the R language, (Ross Ihaka and Robert Gentleman, Auckland, CA, USA) version 4.1.2 (1 November 2021) [101] were used. Box plots were created using the R package ggplot2 v.3.3.6. Principal component analysis (PCA) was performed using the princomp base R function. Descriptive statistics of MT were collected using the describe and describeBy functions from the R psych package v.2.2.5 [102].

The Kruskal-Wallis test was performed using the built-in R function, kruskal.test, while the follow-up Dunn’s test for pairwise comparisons was run using the dunn_test function from the R rstatix package v.0.7.0 [103]. For both the Kruskal-Wallis’ and Dunn’s tests, the seven DTs were treated as dependent variables and the location was used as an explanatory variable. A *p*-value cut-off of 0.05 was used to assess significance and the Benjamini-Hochberg FDR approach [104] was applied for *p*-value adjustment for multiple comparisons in the Dunn’s test.

Finally, a Mantel test based on Pearson’s product-moment correlation was performed to see if there was a significant correlation between the two distance matrices used for generating dendrograms corresponding to the phytochemical data and MTs, respectively. Dendrograms were generated for both the MT data and the biochemical data, by first computing the Euclidean distance matrix using the dist function in R, and then performing hierarchical clustering using the R function hclust and plotting the results using the standard plot function in R. The Mantel test was then performed on the two distance matrices using the mantel function from the R vegan package [105]. GeoTIFF files were acquired from the worldclim online database [106]. Geoclimatic data (minimal temperature, maximal temperature and precipitation) were extracted from GeoTIFF files corresponding to the October, 2017, using the R packages raster v.3.5-21 Raster: Geographic Data Analysis and Modeling, R package version 3.5-21. To generate the distance matrix and perform the hierarchical clustering and Mantel tests, the previously mentioned setup was used.

## 5. Conclusions

Our comparative multidisciplinary study clearly demonstrates that the four eastern European insular populations belong to the *Ilex aquifolium* species. Despite the habitual diversity (natural versus cultivated) and the different geographical locations of the four populations, a molecular analysis proved their affiliation. A phytochemical profile analysis of the leaves distinguished the BG population from the those of RO, HU and SR, while a SEM analysis of descriptive MTs related to leaf and stem indicated three subgroups of morphological diversity: HU-SR, BG and RO. Basic statistical parameters followed by Kruskal-Wallis and Dunn’s tests for pairwise comparisons showed similarity between the populations in HU and SR in most of the analyzed MTs, while in the populations in RO and BG, the values were different compared with the HU-SR subgroup and also between each other (RO-BG). The PCA study showed the four location-specific populations where SR and HU were almost overlapping, while RO and BG were represented as distinct populations in terms of the first two identified PCs.

Looking at the MT-associated phenotypic variability, it seems obvious that the four isolated populations of *Ilex aquifolium* featured differences and similarities whose genetic and environmental dependence require further analysis (i.e., next generation sequencing, metabolomics, liquid chromatography/mass spectrometry, transcriptomics, RNA-seq or epigenomics techniques such as ChIP-seq and ATAC-seq).

Accordingly, the presented data represent an early phase of a comprehensive study intended to run over a longer period of time which will monitor the evolution of phenotypic plasticity, and in particular, seasonal responses to water and light.

## Figures and Tables

**Figure 1 plants-11-02022-f001:**
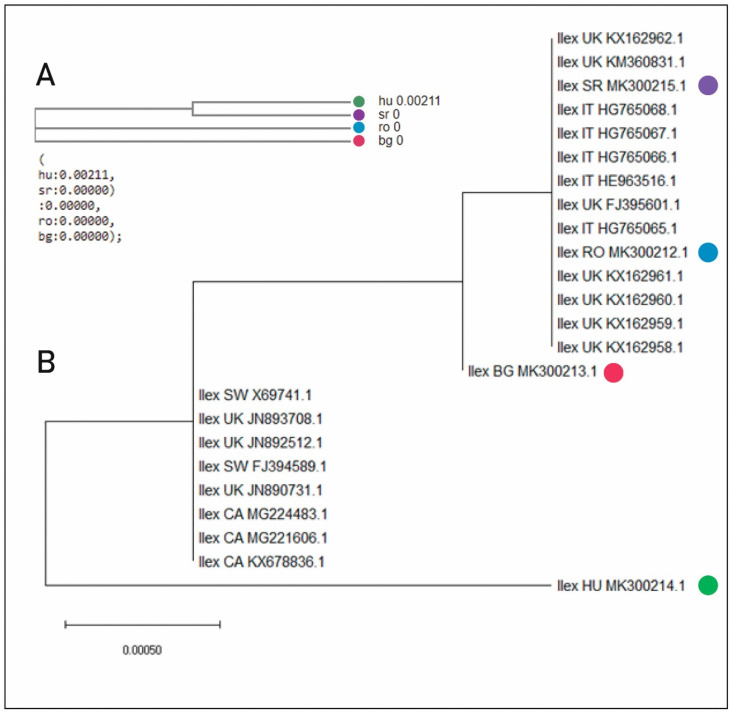
Neighbor-joining tree (**A**) and bootstrap consensus tree (**B**) for *Ilex aquifolium*. (**A**) Multiple alignment of all rbcL four sequences. (**B**) The codes indicate the NCBI accession numbers of the deposited *Ilex aquifolium* specific rbcL sequences, together with the country of origin: RO (blue), HU (green), SR (violet), BG (red), UK (United Kingdom), CA (Canada), SW (Switzerland) and IT (Italy).

**Figure 2 plants-11-02022-f002:**
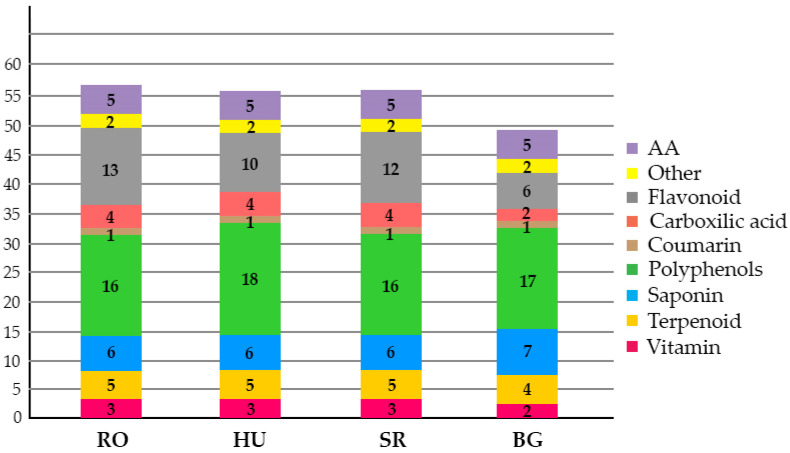
Phytochemical profiles of leaves of all four studied *Ilex aquifolium* populations. The values indicate the abundances of phytochemicals from each category.

**Figure 3 plants-11-02022-f003:**
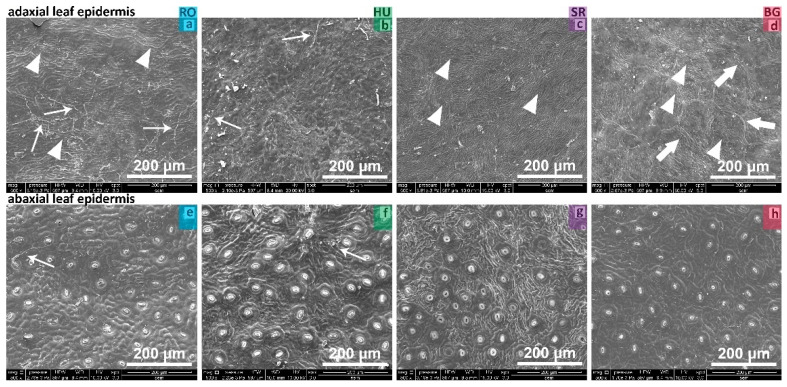
Adaxial and abaxial leaf surfaces, SEM assessment of the morphological aspects. Fine cuticle striations were markedly visible in case of RO, SR and BG samples ((**a**,**c**,**d**) arrow heads); pentagonal or hexagonal shaped cells ((**d**), thick arrows) and fungal hives present on the adaxial and abaxial surfaces of both the HU and RO samples ((**a**,**b**,**e**,**f**), thin arrows), evident stomata presence (**g**,**h**).

**Figure 4 plants-11-02022-f004:**
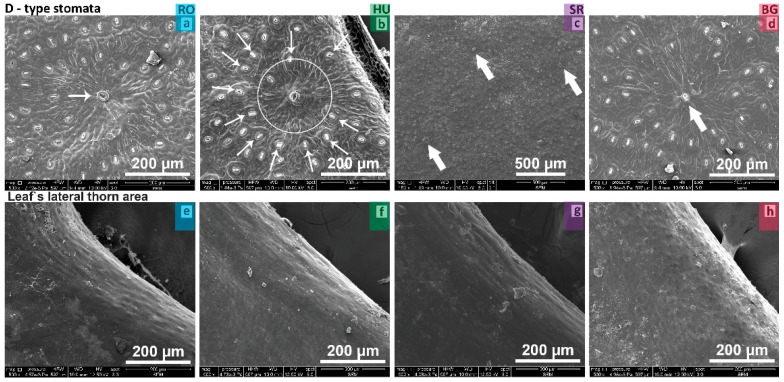
SEM assessment of the morphological aspects: D-type stomata and lateral leaf thorn surface (**e**–**h**). Giant/D-type stomata apparatuses ((**a**,**c**,**d**), thin/thick arrows) and peripheral stomata ((**b**), thin arrows) with radial orientations.

**Figure 5 plants-11-02022-f005:**
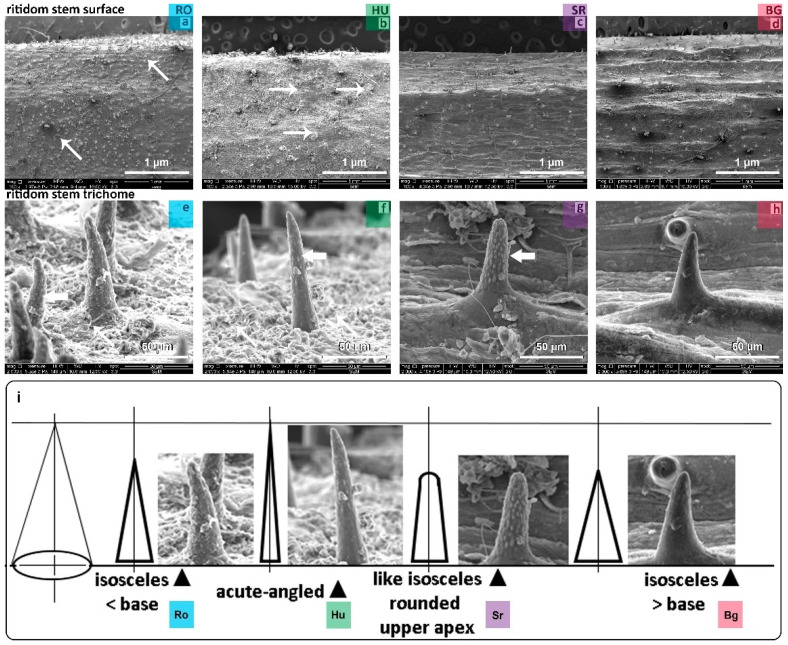
Phenotypic assessment of stem surfaces (**a**–**d**) and trichome morphologies (**e**–**h**). Fungal hives developed deposit structures on RO and HU stem surfaces ((**a**,**b**) thin arrows) and cuticle deposits covering the stem-specific insertion base of stem trichomes in RO and HU (**e**,**f**). Stem trichomes with isosceles triangle-like shapes with different bases (**i**): lower base in RO/HU with an acute-angle apex (**e**,**f**). Stem trichome surfaces—verrucous structures in SR ((**g**), thick arrow), moderate verrucous surface for RO and HU ((**e**,**f**), thick arrow) and underrepresented or totally absent in the BG samples (**h**).

**Figure 6 plants-11-02022-f006:**
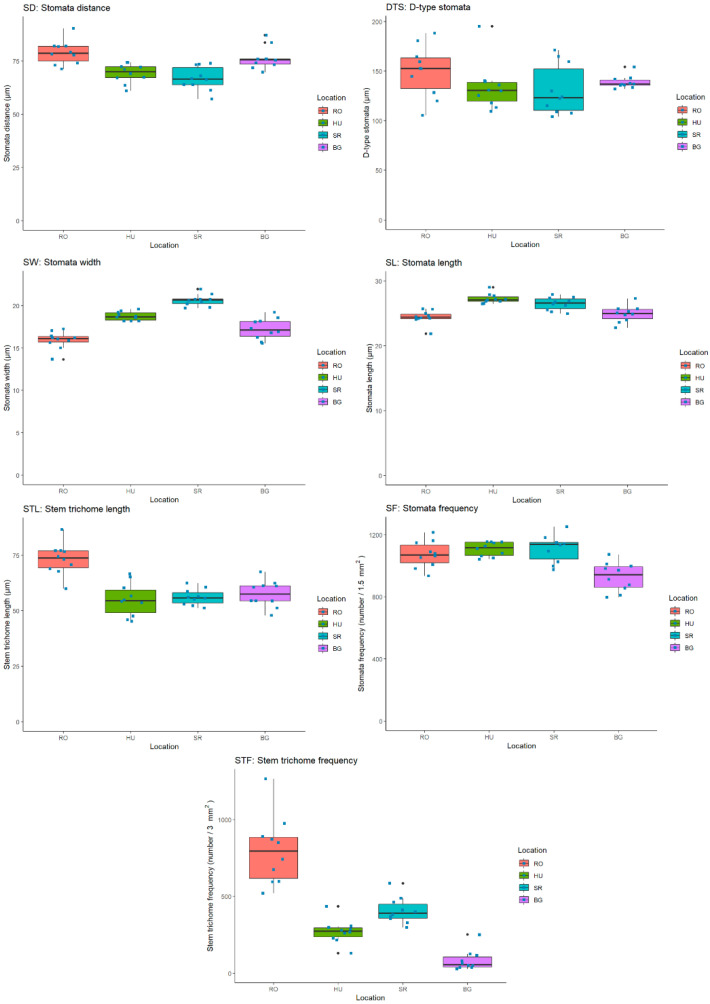
MT-specific median value ranges for all four populations (RO, HU, SR and BG).

**Figure 7 plants-11-02022-f007:**
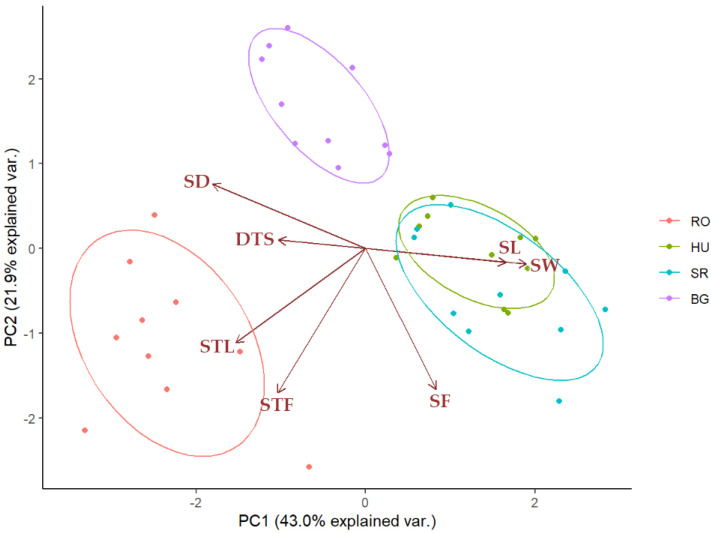
PCA biplot depicting *Ilex aquifolium* population clusters based on the first two principal components, i.e., PC1 and PC2, which accounted for 43.0% and 21.9%, respectively, of the total variation in all populations.

**Figure 8 plants-11-02022-f008:**
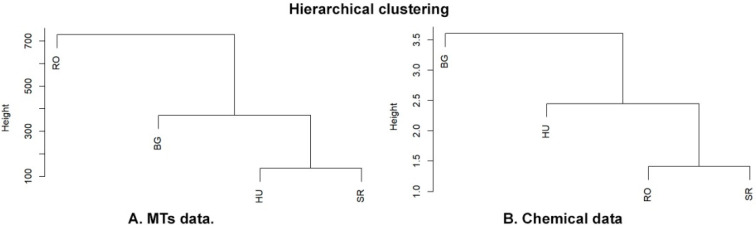
Dendrograms depicting the hierarchical clustering of MT data (**A**) and phytochemical data (**B**), respectively.

**Figure 9 plants-11-02022-f009:**
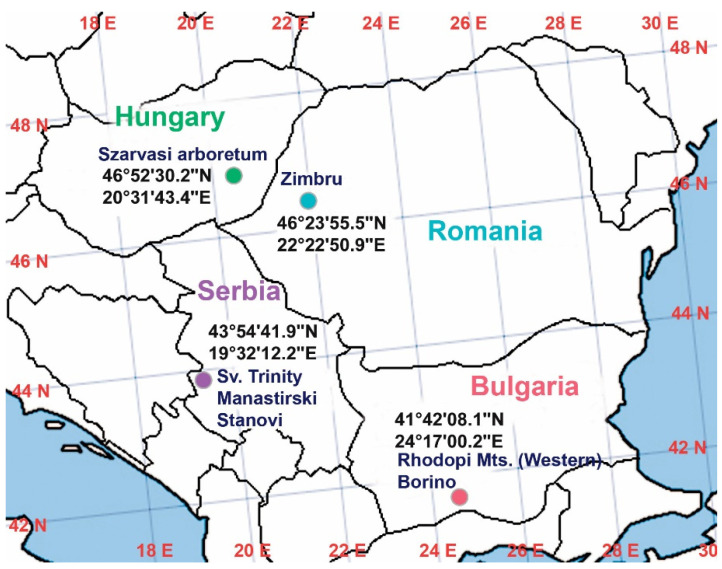
Collection points of *Ilex aquifolium* samples. Romania, Zimbru Reservation, “Dosul Laurului” GPS coordinates: 46°23′55.5″ N, 22°22′50.9″ E; Hungary, Szarvasi arboretum: 46°52′30.2″ N, 20°31′43.4″ E; Bulgaria, Borino, Rhodopi Mts.: 41°42′08.1″ N, 24°17′00.2″ E; and Serbia, middle of the Mala Reka region, Sv. Trinity—Manastirski Stanovi: 43°54′41.9″ N, 19°32′12.2″ E. The color codes on all graphical depictions are as follows: light blue: Romania; green: Hungary; purple: Serbia; and red: Bulgaria.

**Figure 10 plants-11-02022-f010:**
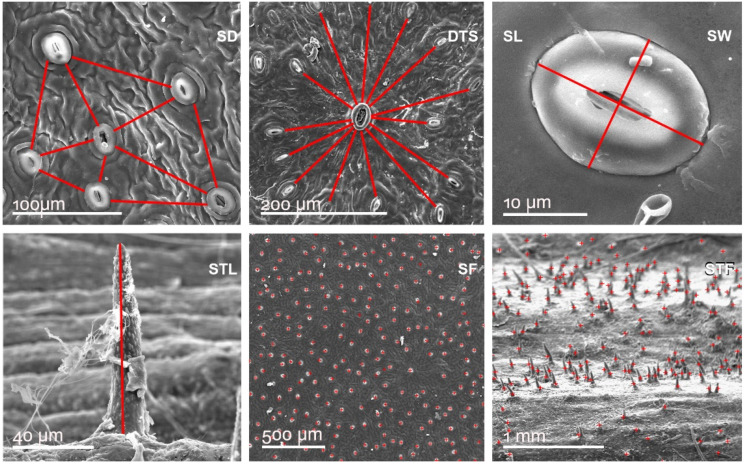
Leaf and stem specific morphological traits (MT) evaluated by morphometry. Leaf stomata distance—SD; Leaf D-type stomata distance—DTS; Leaf stomata length—SL; Leaf stomata width—SW; Stem trichome length—STL; Leaf stomata frequency on 1.5 mm^2^—SF and Stem trichome frequency 3 mm^2^ each—STF.

**Table 1 plants-11-02022-t001:** Phytochemicals identified by UHPLC-ESI-MS analyses in the RO, HU, SR and BG populations.

Category	Name	RO	HU	SR	BG
**Amino acid**	Glutamic acid	+	+	+	+
	Isoleucine/Leucine	+	+	+	+
	Phenylalanine	+	+	+	+
	Tryptophan	+	+	+	+
**Other**	1-Benzofuranecarbaldehyde	+	+	+	+
	Indole-4-carbaldehyde	+	+	+	+
**Flavonoid**	Eriodictyol-O-hexoside	+	−	+	−
	Homoeriodictyol (3′-Methoxy-4′,5,7-trihydroxyflavanone)	+	+	+	+
	Isoquercitrin (Hirsutrin, Quercetin-3-O-glucoside)	+	+	+	−
	Isorhamnetin-3-O-glucoside	+	+	+	+
	Isorhamnetin-3-O-rutinoside (Narcissin)	+	+	+	+
	Kaempferol-O-(rhamnosyl)hexoside	+	+	+	+
	Kaempferol-O-hexoside	+	−	+	−
	Naringenin	+	+	+	+
	Quercetin-3-O-rutinoside-7-O-glucoside	+	+	+	−
	Quercetin-O-(pentosyl)hexoside	+	−	−	−
	Quercetin-O-(rhamnosyl)hexoside-O-hexoside isomer	+	+	+	−
	Reinutrin (Reynoutrin, Quercetin-3-O-xyloside)	+	+	+	−
	Rutin	+	+	+	+
**Carboxylic acid**	12-Oxo phytodienoic acid	+	+	+	−
	Citric acid	+	+	+	+
	Hexadecanedioic acid	+	+	+	−
	Quinic acid	+	+	+	+
**Coumarin**	Dihydroxycoumarin-O-glucoside	+	+	+	+
**Polyphenol**	12-Hydroxyjasmonic acid-12-O-glucoside like Tuberonic acid glucoside	+	+	+	+
	3-O-Feruloylquinic acid	+	+	+	+
	4-O-Feruloylquinic acid	+	+	+	+
	5-O-Feruloylquinic acid	+	+	+	+
	Chlorogenic acid (3-O-Caffeoylquinic acid)	+	+	+	+
	Chryptochlorogenic acid (4-O-Caffeoylquinic acid)	+	+	+	+
	cis-3-O-(4-Coumaroyl)quinic acid	+	+	+	+
	cis-4-O-(4-Coumaroyl)quinic acid	+	+	+	+
	cis-5-O-(4-Coumaroyl)quinic acid	+	+	+	+
	Di-O-caffeoylquinic acid isomer 1	+	+	+	+
	Di-O-caffeoylquinic acid isomer 2	+	+	+	+
	Feruloylquinic acid isomer	−	−	−	+
	Feruloylquinic acid isomer 1	−	+	−	−
	Feruloylquinic acid isomer 2	−	+	−	−
	Neochlorogenic acid (5-O-Caffeoylquinic acid)	+	+	+	+
	trans-3-O-(4-Coumaroyl)quinic acid	+	+	+	+
	trans-4-O-(4-Coumaroyl)quinic acid	+	+	+	+
	trans-5-O-(4-Coumaroyl)quinic acid	+	+	+	+
	trans-Melilotoside (trans-Glucosyl-2-hydroxycinnamate)	+	+	+	+
**Saponin**	Mateglycoside B (Ilexpernoside H, Matenoside A) like Mateglycoside B′	+	+	+	+
	Mateglycoside C like Mateglycoside C’	+	+	+	+
	Mateglycoside D (J3a) like J3b	+	+	+	+
	Matesaponin 1 like Matesaponin 1′	+	+	+	+
	Matesaponin 2 like Matesaponin 2′	+	+	+	+
	Matesaponin 3 (Araliasaponin X) like Matesaponin 3′	−	+	+	+
	Matesaponin 4 like Mateglycoside A (Matenoside C)	+	+	+	+
**Terpenoid**	Abscisic acid	+	+	+	+
	Betulinic acid	+	+	+	+
	Oleanolic acid	+	+	+	+
	Ursolic acid	+	+	+	+
	Uvaol	+	+	+	−
**Vitamin**	Adenine (B4)	+	+	+	+
	Nicotinic acid (Niacin, B3)	+	+	+	+
	Pyridoxine (B6)	+	+	+	−
	Riboflavin (B2)	+	+	+	+

Presence +; Absence −.

**Table 2 plants-11-02022-t002:** Basic statistical parameters (mean, median, sd, range, skew) of MTs in all four populations (RO, HU, SR and BG).

RO											
**Vars**		**n**	**Mean**	**sd**	**Median**	**Trimmed**	**Mad**	**Min**	**Max**	**Range**	**Skew**
**SD**	**1**	10	78.97	5.56	78.69	78.51	4.93	71.25	90.38	19.13	0.45
**DTS**	**2**	10	149.67	26.05	152.60	150.39	26.87	105.34	188.29	82.95	−0.18
**SW**	**3**	10	15.95	1.03	16.14	16.07	0.57	13.67	17.28	3.60	−0.81
**SL**	**4**	10	24.40	1.07	24.40	24.56	0.42	21.85	25.68	3.83	−1.03
**STL**	**5**	10	73.22	7.11	73.76	73.22	4.92	59.87	86.60	26.73	−0.03
**SF**	**6**	10	1073.40	85.57	1071.00	1073.00	103.78	935.00	1215.00	280.00	0.04
**STF**	**7**	10	797.80	221.96	795.50	774.12	223.13	520.00	1265.00	745.00	0.63
**HU**											
**vars**		**n**	**Mean**	**sd**	**Median**	**Trimmed**	**Mad**	**Min**	**Max**	**Range**	**Skew**
**SD**	**1**	10	69.21	4.47	70.05	69.60	4.20	60.98	74.32	13.35	−0.48
**DTS**	**2**	10	133.80	24.02	130.56	129.18	13.79	109.47	195.08	85.61	1.46
**SW**	**3**	10	18.78	0.51	18.70	18.75	0.74	18.20	19.62	1.42	0.23
**SL**	**4**	10	27.26	0.76	27.11	27.14	0.62	26.46	29.00	2.54	1.02
**STL**	**5**	10	55.02	7.48	54.58	54.80	9.35	45.25	66.58	21.33	0.16
**SF**	**6**	10	1108.50	45.55	1119.00	1110.75	53.37	1043.00	1156.00	113.00	−0.26
**STF**	**7**	10	270.40	77.81	273.00	267.50	43.74	129.00	435.00	306.00	0.31
**SR**											
**vars**		**n**	**Mean**	**sd**	**Median**	**Trimmed**	**Mad**	**Min**	**Max**	**Range**	**Skew**
**SD**	**1**	10	66.79	5.57	66.53	67.10	5.90	57.17	73.91	16.74	−0.10
**DTS**	**2**	10	130.76	25.19	123.19	129.03	22.03	104.03	171.33	67.30	0.52
**SW**	**3**	10	20.65	0.68	20.70	20.60	0.55	19.73	21.98	2.26	0.42
**SL**	**4**	10	26.47	1.00	26.59	26.48	1.24	24.94	27.90	2.96	−0.16
**STL**	**5**	10	56.13	3.60	55.85	55.95	4.30	51.24	62.42	11.18	0.30
**SF**	**6**	10	1110.20	87.13	1139.00	1109.38	64.49	975.00	1252.00	277.00	−0.17
**STF**	**7**	10	407.40	84.66	390.00	399.00	71.91	297.00	585.00	288.00	0.70
**BG**											
**vars**		**n**	**Mean**	**sd**	**Median**	**Trimmed**	**Mad**	**Min**	**Max**	**Range**	**Skew**
**SD**	**1**	10	76.28	5.27	75.65	75.76	2.93	69.71	87.07	17.36	0.85
**DTS**	**2**	10	138.78	6.43	136.92	137.70	3.53	131.89	154.29	22.40	1.24
**SW**	**3**	10	17.27	1.24	17.14	17.24	1.48	15.56	19.23	3.67	0.06
**SL**	**4**	10	24.90	1.26	24.96	24.86	1.08	22.78	27.30	4.52	0.12
**STL**	**5**	10	57.51	5.97	57.55	57.47	5.31	47.88	67.50	19.62	0.00
**SF**	**6**	10	929.10	92.30	942.50	927.62	100.08	797.00	1073.00	276.99	−0.05
**STF**	**7**	10	83.49	67.38	56.59	69.50	31.88	27.00	251.99	224.99	1.42
**whole data sets**											
**vars**		**n**	**Mean**	**sd**	**Median**	**Trimmed**	**Mad**	**Min**	**Max**	**Range**	**Skew**
**SD**	**1**	10	72.81	7.12	73.26	72.70	6.46	57.17	90.38	33.21	0.16
**DTS**	**2**	10	138.25	22.33	135.99	136.52	21.97	104.03	195.08	91.05	9.63
**SW**	**3**	10	18.16	1.98	18.20	18.18	2.32	13.67	21.98	8.31	−0.10
**SL**	**4**	10	25.76	1.54	25.68	25.80	1.74	21.85	29.00	7.15	−0.25
**STL**	**5**	10	60.47	9.60	59.26	59.84	8.65	45.25	86.60	41.35	0.67
**SF**	**6**	10	1055.30	107.49	1068.00	1064.41	116.38	797.00	1252.00	455.00	−0.60
**STF**	**7**	10	389.75	293.19	317.50	357.44	283.18	27.00	1265.00	1238.00	0.94

**Table 3 plants-11-02022-t003:** Kruskal-Wallis *p*-values are depicted for each MT, as well as adjusted *p*-values for each of the six pair-wise comparisons (Location L1 vs. Location L2) performed with a Dunn’s test. Significance was considered at a *p*-value threshold of 0.05.

**SD: K-W *p*-val = 0.0001319**							
**No.**	**MT**	**L1**	**L.2**	**Est.1**	**Est.2**	***p* adj.**	***p* adj. signf.**
1	SD	RO	HU	30.5	13.9	0.0038884472	**
2	SD	RO	SR	30.5	10.7	0.0009141059	***
3	SD	RO	BG	30.5	26.9	0.5404890339	ns
4	SD	HU	SR	13.9	10.7	0.5404890339	ns
5	SD	HU	BG	13.9	26.9	0.0193483109	*
6	SD	SR	BG	10.7	26.9	0.0038884472	**
**DTS: K-W *p*-val = 0.1733**							
**No.**	**MT**	**Gr.1**	**Gr.2**	**Est.1**	**Est.2**	***p* adj.**	***p* adj. signf.**
1	DTS	RO	HU	25.9	17.4	0.3119635	ns
2	DTS	RO	SR	25.9	15.7	0.3063532	ns
3	DTS	RO	BG	25.9	23.0	0.6949266	ns
4	DTS	HU	SR	17.4	15.7	0.7450569	ns
5	DTS	HU	BG	17.4	23.0	0.4261672	ns
6	DTS	SR	BG	15.7	23.0	0.3252526	ns
**SW: K-W *p*-val = 4.212 × 10^−7^**							
**No.**	**MT**	**Gr.1**	**Gr.2**	**Est.1**	**Est.2**	***p* adj.**	***p* adj. signf.**
1	SW	RO	HU	7.7	24.5	2.618087 × 10^−3^	**
2	SW	RO	SR	7.7	35.5	6.281448 × 10^−7^	***
3	SW	RO	BG	7.7	14.3	2.067190 × 10^−1^	ns
4	SW	HU	SR	24.5	35.5	5.301561 × 10^−2^	ns
5	SW	HU	BG	24.5	14.3	6.121838 × 10^−2^	ns
6	SW	SR	BG	35.5	14.3	1.499218 × 10^−4^	***
**SL: K-W *p*-val = 3.995 × 10^−5^**							
**No.**	**MT**	**Gr.1**	**Gr.2**	**Est.1**	**Est.2**	***p* adj.**	***p* adj. signf.**
1	SL	RO	HU	9.9	31.9	0.0001545802	***
2	SL	RO	SR	9.9	26.1	0.0038884472	**
3	SL	RO	BG	9.9	14.1	0.4217743968	ns
4	SL	HU	SR	31.9	26.1	0.3207177427	ns
5	SL	HU	BG	31.9	14.1	0.0019873969	**
6	SL	SR	BG	26.1	14.1	0.0325759538	*
**STL: K-W *p*-val = 0.0001979**							
**No.**	**MT**	**Gr.1**	**Gr.2**	**Est.1**	**Est.2**	***p* adj.**	***p* adj. signf.**
1	SL	RO	HU	34.5	13.9	0.0004884332	***
2	SL	RO	SR	34.5	15.8	0.0010434569	**
3	SL	RO	BG	34.5	17.8	0.0028038024	**
4	SL	HU	SR	13.9	15.8	0.7162921150	ns
5	SL	HU	BG	13.9	17.8	0.6835330598	ns
6	SL	SR	BG	15.8	17.8	0.7162921150	ns
**SF: K-W *p*-val = 0.0009187**							
**No.**	**MT**	**Gr.1**	**Gr.2**	**Est.1**	**Est.2**	***p* adj.**	***p* adj. signf.**
1	SF	RO	HU	22.00	26.45	0.574900837	ns
2	SF	RO	SR	22.00	25.70	0.574900837	ns
3	SF	RO	BG	22.00	7.85	0.013588382	*
4	SF	HU	SR	26.45	25.70	0.885920409	ns
5	SF	HU	BG	26.45	7.85	0.001916669	**
6	SF	SR	BG	25.70	7.85	0.001916669	**
**STF: K-W *p*-val = 1.367 × 10^−7^**							
**No.**	**MT**	**Gr.1**	**Gr.2**	**Est.1**	**Est.2**	***p* adj.**	***p* adj. signf.**
1	SF	RO	HU	35.4	16.1	6.005562 × 10^−4^	***
2	SF	RO	SR	35.4	24.7	5.859064 × 10^−2^	ns
3	SF	RO	BG	35.4	5.8	8.993826 × 10^−8^	***
4	SF	HU	SR	16.1	24.7	9.998055 × 10^−2^	ns
5	SF	HU	BG	16.1	5.8	5.859064 × 10^−2^	ns
6	SF	SR	BG	24.7	5.8	6.005562 × 10^−4^	***

Significance codes: *p*. adj—adjusted *p*-value; *p*. adj. signif.—significance level for *p*. adj. *p*. adj significance intervals—0 *** 0.001 ** 0.01 * 0.05. 0.1 ns 1.

**Table 4 plants-11-02022-t004:** Climatic conditions and altitudes for *Ilex aquifolium* population sampling areas.

	Zimbru/RO			Szarvas/HU			Mala-Reka/SR			Borino/BG		
**Min/Avg/Max** **temp. (°C)—October**	6	**13**	17	10	**15**	18	10	**15**	18	7	**10**	13
**Min/Avg/Max** **temp. (°C)—2017**	−11	**14**	32	−9	**14**	32	−7	14	32	−11	**9.6**	24
**Avg. humidity(%)—October**	69			62			63			70		
**Min/Avg/Max humidity(%)—2017**	52	**74**	90	43	**67**	83	50	**68**	86	56	**75**	91
**Avg. rainfall amount(mm)—October**	67.1			76.9			63			123.3		
**Min/Avg/Max rainfall amount(mm)—2017**	23.2	**66.6**	160.1	42.3	**67.5**	114.6	19.9	**71**	132.7	70.3	**155**	333
**Min/Avg/Max** **rainy days/month—2017**	1	**6**	12	2	**4.9**	7	1	**5.8**	13	4	**10**	24
**Altitude(m)**	258			79			1033			1143		

Climatic variables were extracted from World Weather Online in accordance with the collection points (https://www.worldweatheronline.com/, accessed on 19 June 2022).

## Data Availability

Links to publicly archived datasets analyzed: Seq1Ro MK300212—*Ilex aquifolium* voucher IlxRo ribulose-1,5-bisphosphate carboxylase/oxygenase large subunit (rbcL) gene, partial cds; chloroplast https://www.ncbi.nlm.nih.gov/nuccore/1699097326 (accessed on 2 February, 2020); Seq3Hu MK300214*—Ilex aquifolium* voucher IlxHu ribulose-1,5-bisphosphate carboxylase/oxygenase large subunit (rbcL) gene, partial cds; chloroplast https://www.ncbi.nlm.nih.gov/nuccore/1699097330 (accessed on 2 February 2020); Seq4Sr MK300215 *—Ilex aquifolium* voucher IlxSr ribulose-1,5-bisphosphate carboxylase/oxygenase large subunit (rbcL) gene, partial cds; chloroplast https://www.ncbi.nlm.nih.gov/nuccore/1699097332 (accessed on 2 February 2020); Seq2Bg MK300213*—Ilex aquifolium* voucher IlxBg ribulose-1,5-bisphosphate carboxylase/oxygenase large subunit (rbcL) gene, partial cds; chloroplast https://www.ncbi.nlm.nih.gov/nuccore/1699097328 (accessed on 2 February 2020).

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
