# Peer review of "Assessing Phenotypic Variability in Some Eastern European Insular Populations of the Climatic Relict Ilex aquifolium L."

_plants, 2022, doi:10.3390/plants11152022_

Round 1

Reviewer 1 Report

Dear Editor and Authors,

The paper analyzes population diversity of the common holly from four insular populations in Eastern Europe (Romania, Hungary, Serbia and Bulgaria). The aim of the study was to perform a comparative analysis of descriptive and quantitative morphological traits, phytonutrient profiling and phylogenetic analysis. I consider it particularly valuable that the authors cover phenotype from two aspects: morphometric and chemical. Although the study includes only a smaller number of populations and individuals, it is valuable as it pertains to an endangered species which is on the IUCN Red List. All in all, I do think that the paper could be interesting for publication in the Plants journal, but before that some things need to be clarified and better explained. Therefore, I suggest a major revision.

Although I personally find the first and second paragraphs to be informative and interesting, I do not think they should be in this paper. Namely, this paper is not a true phylogenetic study that covers a larger number of different species from the genus Ilex. Instead I would expect more information on the studied species and the issue being studied. Those two paragraphs should be abbreviated and revised significantly. Furthermore, I would be particularly keen to know why the authors have focused on the rbcL gene, which is considered a generic plastid marker? This should be explained better in the introduction. It is interesting that the authors have samples both from natural populations and from an arboretum. It is well known that arboreta have collections of woody plants from all over the world, hence in that case the above marker could be suitable to detect species or even hybrids between the studied species and some other in the arboretum, all of the same genus. All in all, the study hypotheses should be defined better and this should be presented in an interesting and clearer way. The paper is quite extensive so in some parts, and this applies to the introduction in particular, it is hard to determine what issue exactly is being studied.

I do not see why the studied species on the territory of Romania has been presented with a special paragraph. The entire study area should be presented together in a special paragraph. In addition, I would also like to read something general on the comparative analysis of descriptive and quantitative morphological traits, phytonutrient profiling and phylogenetic analysis.

The results from Figure 1 should be described more clearly.

I do not consider subsection 2.2. Habitat and plant association as results. This part of the manuscript can be used to write the introductory part of the paper.

Figure 2 can be moved to Supplemental materials.

In some places the authors use the term population, in some locality. This should be harmonized throughout the paper.

I think that the tables from Supplemental materials should be enclosed as real tables in Word document, not as tif photograph.

Figure 8 should be of better quality. The circle with the factor loadings can be erased. It does not contribute to the clarity of the figure.

Although the discussion is written clearly and reads fluently, certain parts boil down to a repetition of the results. The results could be discussed better. In MDPI journals in the field of plant science, such as e.g. Plants, Diversity and Forests, a number of papers on morphometric and chemical analyses has been published that could be used in your paper. I would once again like to highlight that a particular value of this study is the combination of morphometric and chemical analyses, and this should in a way also be emphasized in the discussion.

Furthermore, the authors highlight in several places the relict character of the studied species, even mentioning some other species such as common yew. The discussion could address the broader readership and highlight the values of the Balkan Peninsula in terms of diversity. Here you could also mention more recent studies, similar to your paper, for other woody species.

Overall, the results of this study should be compared more to the results of more recent studies. More references should be included in the discussion pertaining to morphometric and chemical studies, plasticity and adaptability.

In the results I would also like to see a correlation analysis that would involve environmental data. The authors could also perform a Mantel test.

The basic assumption for running ANOVA is homogeneity of variance.

Minor corrections:

There is no need to provide the name of the family in the title.

Line 21 and 37 – Ilex aquifolium L.

Authorities are not written with the full name as the authors have done, but rather with standardized abbreviation. The same applies to other botanical names mentioned in the paper. With the scientific name for each species mentioned in the paper for the first time you also need to specify authorities.

The scientific names of species and genera are always written in italics.

Reviewer 2 Report

See attached PDF.

Reviewer 3 Report

The presented manuscript concentrates on a multidisciplinary study of the analysis of descriptive and quantitative morphological traits, phytochemicals profile, and phylogenetic analysis of individuals belonging to insular populations of Ilex aquifolium located in four different countries. This comparative and broad approach acertaines presented work's novelty and scientific soundness. The manuscript seems suitable for publication in the Plants journal, nevertheless, some remarks can ae addressed to the authors. 

  1. "Phytonutrients: term used in 2.3 subsections a little bit misleading. Saponins from this plant are supposed to have a toxic effect in high doses. This is why Ilex aqafolium usage is limited in contemporary herbalism. I would consider term "phytochemicals" more appropriate. 
  2. It would be valuable for the readers and for the scientist performing chemotaxonomic analysis to have and insight into UHPLC-MS chromatograms obtained during the compounds analysis to compare their results. Such chromatograms could be included into Supplementary files. Also, I am not able as reviewer thoroughly evaluate the current chemical analysis without details. Did the Authors identify only known compounds or find the new ones based on structural analysis? Were the described compounds only dominant, or were they selected as a known, previously identified by other authors? 

  1. Line 590 . What was the method of the compound identification? Was it conducted based on authentic standards analysis or by comparing with particular library data? What was the solvent gradient? What was the ion detection mode, positive or negative? Please provide the details to meet the Plants Material and Methods requirements: “They should be described with sufficient detail to allow others to replicate and build on published results.”

  1. Line 408 profile not profilea

Round 2

Reviewer 1 Report

I have read the reviewed version of the paper with great interest. Nevertheless, there are still some things that could be improved.

There is little research that deals with both morphological and chemical properties, and that is what adds value to this paper. The paper needs to include more recent literature tackling similar research – chemical and morphological variability of plants in the Discussion section. In this way the authors will address a broader readership. In MDPI journals in the field of plant science, such as e.g. Plants, Diversity and Forests, a number of papers on morphometric and chemical analyses has been published that could be used in your paper.

Furthermore, it is possible to carry out the Mantel test between morpho/chem and geographical (IBD pattern) and environmental (IBE pattern) variables. Moreover, the authors stated that one of the goals in this study was: “to set the grounds for a long-term study on the above mentioned four populations of Ilex aquifolium in order to assess their adaptability and acclimation potential with respect to the evolution of Eastern European specific climatic conditions”. However, environmental variables were not analysed in the research. BioClim variables can be included in the research (https://www.worldclim.org/data/bioclim.html).

Minor corrections:

Line 19 – L. not in italics

 Line 76 – A. austriacum

Lines 401 and 402 – single quotation marks: ‘Yerba Mat’, ‘Argentea Mariginata’, ‘Blue Angelwere’ Line 402 - Ilex × meserveae - multiplication sign (×), not the letter x; × not in italics Lines 1133 – 1136 - Ilex genus: I. pubescens, I. macropoda etc.

Line 1136 – write species names in italics

Reviewer 2 Report

See attached PDF.

Round 3

Reviewer 1 Report

Dear authors and editor,

I have read the reviewed version of the paper with great interest. In my opinion, the paper has been improved significantly, and a few more comments and minor corrections are provided below.

All the best in further work!

Line 35 L. not in italics

Line 41 - Ilex-Taxus in italics

Line 44 Ilex L.

Lines 57 and 60 instead former Yugoslavian republics write the names of the countries

Line 66 A. pseudoplatanus L., A. austriacum Tratt.

Line 67 instead Quercus genera please write Quercus species

Lines 82 and 83 instead with Betulaceae spp. and Quercus spp., please write with species of Betulaceae family and Quercus genus

Line 155 - ‛“Blue Angelwere”′ please write Blue Angel were

Line 605 Amaranthus L.

Line 799 Ilex in italics
